# Collaborative Deep Learning in Fixed Topology Networks

Zhanhong Jiang[1], Aditya Balu[1], Chinmay Hegde[2], and Soumik Sarkar[1]

[1]Department of Mechanical Engineering, Iowa State University,
$zhjiang, baditya, soumiks$@iastate.edu
[2]Department of Electrical and Computer Engineering , Iowa State University, $chinmay$@iastate.edu

## Abstract

There is significant recent interest to parallelize deep learning algorithms in order to handle the enormous growth in data and model sizes. While most advances focus on model parallelization and engaging multiple computing agents via using a central parameter server, aspect of data parallelization along with decentralized computation has not been explored sufficiently. In this context, this paper presents a new consensus-based distributed SGD (CDSGD) (and its momentum variant, CDMSGD) algorithm for collaborative deep learning over fixed topology networks that enables data parallelization as well as decentralized computation. Such a framework can be extremely useful for learning agents with access to only local/private data in a communication constrained environment. We analyze the convergence properties of the proposed algorithm with strongly convex and nonconvex objective functions with fixed and diminishing step sizes using concepts of Lyapunov function construction. We demonstrate the efficacy of our algorithms in comparison with the baseline centralized SGD and the recently proposed federated averaging algorithm (that also enables data parallelism) based on benchmark datasets such as MNIST, CIFAR-10 and CIFAR-100.

## 1 Introduction

In this paper, we address the scalability of optimization algorithms for deep learning in a distributed setting. Scaling up deep learning [1] is becoming increasingly crucial for large-scale applications where the sizes of both the available data as well as the models are massive [2]. Among various algorithmic advances, many recent attempts have been made to parallelize stochastic gradient descent (SGD) based learning schemes across multiple computing agents. An early approach called Downpour SGD [3], developed within Google's *disbelief* software framework, primarily focuses on model parallelization (i.e., splitting the model across the agents). A different approach known as elastic averaging SGD (EASGD) [4] attempts to improve perform multiple SGDs in parallel; this method uses a central parameter server that helps in assimilating parameter updates from the computing agents. However, none of the above approaches concretely address the issue of *data* parallelization, which is an important issue for several learning scenarios: for example, data parallelization enables privacy-preserving learning in scenarios such as distributed learning with a network of mobile and Internet-of-Things (IoT) devices. A recent scheme called *Federated Averaging SGD* [5] attempts such a data parallelization in the context of deep learning with significant success; however, they still use a central parameter server.

In contrast, deep learning with decentralized computation can be achieved via gossip SGD algorithms [6, 7], where agents communicate probabilistically without the aid of a parameter server. However, decentralized computation in the sense of gossip SGD is not feasible in many real life applications. For instance, consider a large (wide-area) sensor network [8, 9] or multi-agent robotic

Table 1: Comparisons between different optimization approaches

| Method | $f$ | $\nabla f$ | Step Size | Con.Rate | D.P. | D.C. | C.C.T. |
|---|---|---|---|---|---|---|---|
| SGD | Str-con | Lip. | Con. | $\mathcal{O}(\gamma^k)$ | No | No | No |
| Downpour SGD [3] | Nonconvex | Lip. | Con.&Ada. | N/A | Yes | No | No |
| EASGD [4] | Str-con | Lip. | Con. | $\mathcal{O}(\gamma^k)$ | No | No | No |
| Gossip SGD [7] | Str-con | Lip.&Bou. | Con. | $\mathcal{O}(\gamma^k)$ | No | Yes | No |
|  | Str-con | Lip.&Bou. | Dim. | $\mathcal{O}(\frac{1}{k})$ |  |  |  |
| FedAvg [5] | Nonconvex | Lip. | Con. | N/A | Yes | No | No |
| CDSGD [This paper] | Str-con | Lip.&Bou. | Con. | $\mathcal{O}(\gamma^k)$ | Yes | Yes | Yes |
|  | Str-con | Lip.&Bou. | Dim. | $\mathcal{O}(\frac{1}{k^\epsilon})$ |  |  |  |
|  | Nonconvex | Lip.&Bou. | Con. | N/A |  |  |  |
|  | Nonconvex | Lip.&Bou. | Dim. | N/A |  |  |  |

Con.Rate: convergence rate, Str-con: strongly convex. Lip.&Bou.: Lipschitz continuous and bounded. Con.: constant and Con.&Ada.: constant&adagrad. Dim.: diminishing. $\gamma \in (0,1)$ is a positive constant. $\epsilon \in (0.5, 1]$ is a positive constant. D.P.: data parallelism. D.C.: decentralized computation. C.C.T.: constrained communication topology.

network that aims to learn a model of the environment in a collaborative manner [10, 11]. For such cases, it may be infeasible for arbitrary pairs of agents to communicate on-demand; typically, agents are only able to communicate with their respective neighbors in a communication network in a fixed (or evolving) topology.

**Contribution**: This paper introduces a new class of approaches for deep learning that enables both data parallelization and decentralized computation. Specifically, we propose consensus-based distributed SGD (CDSGD) and consensus-based distributed momentum SGD (CDMSGD) algorithms for collaborative deep learning that, for the first time, satisfies all three requirements: data parallelization, decentralized computation, and constrained communication over fixed topology networks. Moreover, while most existing studies solely rely on empirical evidence from simulations, we present rigorous convergence analysis for both (strongly) convex and non-convex objective functions, with both fixed and diminishing step sizes using a Lyapunov function construction approach. Our analysis reveals several advantages of our method: we match the best existing rates of convergence in the centralized setting, while simultaneously supporting data parallelism as well as constrained communication topologies; to our knowledge, this is the first approach that achieves all three desirable properties; see Table 1 for a detailed comparison.

Finally, we validate our algorithms' performance on benchmark datasets, such as MNIST, CIFAR-10, and CIFAR-100. Apart from centralized SGD as a baseline, we also compare performance with that of Federated Averaging SGD as it also enables data parallelization. Empirical evidence (for a given number of agents and other hyperparametric conditions) suggests that while our method is slightly slower, we can achieve higher accuracy compared to the best available algorithm (Federated Averaging (FedAvg)). Empirically, the proposed framework in this work is suitable for situations without central parameter servers, but also robust to a central parameter server failture situation.

**Related work**: Apart from the algorithms mentioned above, a few other related works exist, including a distributed system called Adam for large deep neural network (DNN) models [12] and a distributed methodology by Strom [13] for DNN training by controlling the rate of weight-update to reduce the amount of communication. Natural Gradient Stochastic Gradient Descent (NG-SGD) based on model averaging [14] and staleness-aware async-SGD [15] have also been developed for distributed deep learning. A method called CentralVR [16] was proposed for reducing the variance and conducting parallel execution with linear convergence rate. Moreover, a decentralized algorithm based on gossip protocol called the multi-step dual accelerated (MSDA) [17] was developed for solving deterministically smooth and strongly convex distributed optimization problems in networks with a provable optimal linear convergence rate. A new class of decentralized primal-dual methods [18] was also proposed recently in order to improve inter-node communication efficiency for distributed convex optimization problems. To minimize a finite sum of nonconvex functions over a network, the authors in [19] proposed a zeroth-order distributed algorithm (ZENITH) that was globally convergent with a sublinear rate. From the perspective of distributed optimization, the proposed algorithms have similarities with the approaches of [20, 21]. However, we distinguish our work due to the collaborative learning aspect with data parallelization and extension to the stochastic setting and nonconvex objective functions. In [20] the authors only considered convex objective functions in a

deterministic setting, while the authors in [21] presented results for non-convex optimization problems in a deterministic setting. Our proof techniques are different from those in [20, 21] with the choice of Lyapunov function, as well as the notion of stochastic Lyapunov gradient. More importantly, we provide an extensive and thorough suite of numerical comparisons with both centralized methods and distributed methods on benchmark datasets.

The rest of the paper is organized as follows. While section 2 formulates the distributed, unconstrained stochastic optimization problem, section 3 presents the CDSGD algorithm and the Lyapunov stochastic gradient required for analysis presented in section 4. Validation experiments and performance comparison results are described in section 5. The paper is summarized, concluded in section 6 along with future research directions. Detailed proofs of analytical results, extensions (e.g., effect of diminishing step size) and additional experiments are included in the supplementary section 7.

## 2 Formulation

We consider the standard (unconstrained) empirical risk minimization problem typically used in machine learning problems (such as deep learning):

$$\min \frac{1}{n} \sum_{i=1}^{n} f^i(x), \tag{1}$$

where $x \in \mathbb{R}^d$ denotes the parameter of interest and $f : \mathbb{R}^d \to \mathbb{R}$ is a given loss function, and $f^i$ is the function value corresponding to a data point $i$. In this paper, we are interested in learning problems where the computational agents exhibit *data parallelism*, i.e., they only have access to their own respective training datasets. However, we assume that the agents can communicate over a static undirected graph $\mathcal{G} = (\mathcal{V}, \mathcal{E})$, where $\mathcal{V}$ is a vertex set (with nodes corresponding to agents) and $\mathcal{E}$ is an edge set. With $N$ agents, we have $\mathcal{V} = \{1, 2, ..., N\}$ and $\mathcal{E} \subseteq \mathcal{V} \times \mathcal{V}$. If $(j, l) \in \mathcal{E}$, then Agent $j$ can communicate with Agent $l$. The neighborhood of agent $j \in \mathcal{V}$ is defined as: $Nb(j) \triangleq \{l \in \mathcal{V} : (j, l) \in \mathcal{E} \text{ or } j = l\}$. Throughout this paper we assume that the graph $\mathcal{G}$ is *connected*. Let $\mathcal{D}_j$, $j = 1, \ldots, n$ denote the subset of the training data (comprising $n_j$ samples) corresponding to the $j^{\text{th}}$ agents such that $\sum_{j=1}^{N} n_j = n$. With this setup, we have the following simplification of Eq. 1:

$$\min \frac{1}{n} \sum_{j=1}^{N} \sum_{i \in \mathcal{D}_j} f^i(x) = \frac{N}{n} \sum_{j=1}^{N} \sum_{i \in \mathcal{D}_j} f_j^i(x), \tag{2}$$

where, $f_j(x) = \frac{1}{N} f(x)$ is the objective function specific to Agent $j$. This formulation enables us to state the optimization problem in a distributed manner, where $f(x) = \sum_{j=1}^{N} f_j(x)$. [1] Furthermore, the problem (1) can be reformulated as

$$\min \frac{N}{n} \mathbf{1}^T \mathbf{F}(\mathbf{x}) := \frac{N}{n} \sum_{j=1}^{N} \sum_{i \in \mathcal{D}_j} f_j^i(x^j) \tag{3a}$$

$$\text{s.t. } x^j = x^l \ \forall (j, l) \in \mathcal{E}, \tag{3b}$$

where $\mathbf{x} := (x^1, x^2, \ldots, x^N)^T \in \mathbb{R}^{N \times d}$ and $\mathbf{F}(\mathbf{x})$ can be written as

$$\mathbf{F}(\mathbf{x}) = \left[ \sum_{i \in \mathcal{D}_1} f_1^i(x^1), \ \sum_{i \in \mathcal{D}_2} f_2^i(x^2), \ldots, \ \sum_{i \in \mathcal{D}_N} f_N^i(x^N) \right]^T \tag{4}$$

Note that with $d > 1$, the parameter set $\mathbf{x}$ as well as the gradient $\nabla \mathbf{F}(\mathbf{x})$ correspond to matrix variables. However, for simplicity in presenting our analysis, we set $d = 1$ in this paper, which corresponds to the case where $\mathbf{x}$ and $\nabla \mathbf{F}(\mathbf{x})$ are vectors.

We now introduce several key definitions and assumptions that characterize the objective functions and the agent interaction matrix.

**Definition 1.** *A function $f : \mathbb{R}^d \to \mathbb{R}$ is H-strongly convex, if for all $x, y \in \mathbb{R}^d$, we have $f(y) \geq f(x) + \nabla f(x)^T (y - x) + \frac{H}{2}\|y - x\|^2$.*

**Definition 2.** *A function $f : \mathbb{R}^d \to \mathbb{R}$ is $\gamma$-smooth if for all $x, y \in \mathbb{R}^d$, we have $f(y) \leq f(x) + \nabla f(x)^T (y - x) + \frac{\gamma}{2}\|y - x\|^2$.*

As a consequence of Definition 2, we can conclude that $\nabla f$ is Lipschitz continuous, i.e., $\|\nabla f(y) - \nabla f(x)\| \leq \gamma \|y - x\|$ [22].

**Definition 3.** *A function $c$ is said to be coercive if it satisfies: $c(x) \to \infty$ $when \|x\| \to \infty$.*

**Assumption 1.** *The objective functions $f_j : \mathbb{R}^d \to \mathbb{R}$ are assumed to satisfy the following conditions: a) Each $f_j$ is $\gamma_j$-smooth; b) each $f_j$ is proper (not everywhere infinite) and coercive; and c) each $f_j$ is $L_j$-Lipschitz continuous, i.e., $|f_j(y) - f_j(x)| < L_j \|y - x\|$ $\forall x, y \in \mathbb{R}^d$.*

As a consequence of Assumption 1, we can conclude that $\sum_{j=1}^{N} f_j(x^j)$ possesses Lipschitz continuous gradient with parameter $\gamma_m := \max_j \gamma_j$. Similarly, each $f_j$ is strongly convex with $H_j$ such that $\sum_{j=1}^{N} f_j(x^j)$ is strongly convex with $H_m = \min_j H_j$.

Regarding the communication network, we use $\Pi$ to denote the agent interaction matrix, where the element $\pi_{jl}$ signifies the link weight between agents $j$ and $l$.

**Assumption 2.** *a) If $(j, l) \notin \mathcal{E}$, then $\pi_{jl} = 0$; b) $\Pi^T = \Pi$; c) $null\{I - \Pi\} = span\{\mathbf{1}\}$; and d) $I \succeq \Pi \succ -I$.*

The main outcome of Assumption 2 is that the probability transition matrix is doubly stochastic and that we have $\lambda_1(\Pi) = 1 > \lambda_2(\Pi) \geq \cdots \geq \lambda_N(\Pi) \geq 0$, where $\lambda_z(\Pi)$ denotes the $z$-th largest eigenvalue of $\Pi$.

## 3 Proposed Algorithm

### 3.1 Consensus Distributed SGD

For solving stochastic optimization problems, SGD and its variants have been commonly used to centralized and distributed problem formulations. Therefore, the following algorithm is proposed based on SGD and the concept of consensus to solve the problem laid out in Eq. 2,

$$x_{k+1}^j = \sum_{l \in Nb(j)} \pi_{jl} x_k^l - \alpha g_j(x_k^j) \tag{5}$$

where $Nb(j)$ indicates the neighborhood of agent $j$, $\alpha$ is the step size, $g_j(x_k^j)$ is stochastic gradient of $f_j$ at $x_k^j$, which corresponds to a minibatch of sampled data points at the $k^{th}$ epoch. More formally, $g_j(x_k^j) = \frac{1}{b'} \sum_{q' \in \mathcal{D}'} \nabla f_j^{q'}(x_k^j)$, where $b'$ is the size of the minibatch $\mathcal{D}'$ randomly selected from the data subset $\mathcal{D}_j$. While the pseudo-code of CDSGD is shown below in Algorithm 1, momentum versions of CDSGD based on Polyak momentum [23] and Nesterov momentum [24] are also presented in the supplementary section 7. In experiments, Nesterov momentum is used as it has been shown in the traditional SGD implementations that the Nesterov variant outperforms the Polyak momentum. Note, that mini-batch implementations of these algorithms are straightforward, hence,

are not discussed here in detail, and that the convergence analysis of momentum variants is out of scope in this paper and will be presented in our future work.

---

**Algorithm 1:** CDSGD

---

**Input** : $m, \alpha, N$
**Initialize:** $x_0^j, (j = 1, 2, \ldots, N)$
Distribute the training dataset to $N$ agents.
**for** *each agent* **do**
    **for** $k = 0 : m$ **do**
        Randomly shuffle the corresponding data subset $\mathcal{D}_j$ (without replacement)
        $w_{k+1}^j = \sum_{l \in Nb(j)} \pi_{jl} x_k^l$
        $x_{k+1}^j = w_{k+1}^j - \alpha g_j(x_k^j)$
    **end**
**end**

---

### 3.2 Tools for convergence analysis

We now analyze the convergence properties of the iterates $\{x_k^j\}$ generated by Algorithm 1. The following section summarizes some key intermediate concepts required to establish our main results.

First, we construct an appropriate *Lyapunov* function that will enable us to establish convergence. Observe that the update law in Alg. 1 can be expressed as:

$$\mathbf{x}_{k+1} = \Pi \mathbf{x}_k - \alpha \mathbf{g}(\mathbf{x}_k), \tag{6}$$

where

$$\mathbf{g}(\mathbf{x}_k) = [g_1(x_k^1) g_2(x_k^2) ... g_N(x_k^N)]^T$$

Denoting $\mathbf{w}_k = \Pi \mathbf{x}_k$, the update law can be re-written as $\mathbf{x}_{k+1} = \mathbf{w}_k - \alpha \mathbf{g}(\mathbf{x}_k)$. Moreover, $\mathbf{x}_{k+1} = \mathbf{x}_k - \mathbf{x}_k + \mathbf{w}_k - \alpha \mathbf{g}(\mathbf{x}_k)$. Rearranging the last equality yields the following relation:

$$\mathbf{x}_{k+1} = \mathbf{x}_k - \alpha(\mathbf{g}(\mathbf{x}_k) + \alpha^{-1}(\mathbf{x}_k - \mathbf{w}_k)) = \mathbf{x}_k - \alpha(\mathbf{g}(\mathbf{x}_k) + \alpha^{-1}(I - \Pi)\mathbf{x}_k) \tag{7}$$

where the last term in Eq. 7 is the *Stochastic Lyapunov Gradient*. From Eq. 7, we observe that the "effective" gradient step is given by $\mathbf{g}(\mathbf{x}_k) + \alpha^{-1}(I - \Pi)\mathbf{x}_k$. Rewriting $\nabla \mathcal{J}^i(\mathbf{x}_k) = \mathbf{g}(\mathbf{x}_k) + \alpha^{-1}(I - \Pi)\mathbf{x}_k$, the updates of CDSGD can be expressed as:

$$\mathbf{x}_{k+1} = \mathbf{x}_k - \alpha \nabla \mathcal{J}^i(\mathbf{x}_k). \tag{8}$$

The above expression naturally motivates the following Lyapunov function candidate:

$$V(\mathbf{x}, \alpha) := \frac{N}{n} \mathbf{1}^T \mathbf{F}(\mathbf{x}) + \frac{1}{2\alpha} \|\mathbf{x}\|_{I-\Pi}^2 \tag{9}$$

where $\| \cdot \|_{I-\Pi}$ denotes the norm with respect to the PSD matrix $I - \Pi$. Since $\sum_{j=1}^N f_j(x^j)$ has a $\gamma_m$-Lipschitz continuous gradient, $\nabla V(\mathbf{x})$ also is a Lipschitz continuous gradient with parameter:

$$\hat{\gamma} := \gamma_m + \alpha^{-1}\lambda_{\max}(I - \Pi) = \gamma_m + \alpha^{-1}(1 - \lambda_N(\Pi)).$$

Similarly, as $\sum_{j=1}^N f_j(x^j)$ is $H_m$-strongly convex, then $V(\mathbf{x})$ is strongly convex with parameter:

$$\hat{H} := H_m + (2\alpha)^{-1}\lambda_{\min}(I - \Pi) = H_m + (2\alpha)^{-1}(1 - \lambda_2(\Pi)).$$

Based on Definition 1, $V$ has a unique minimizer, denoted by $\mathbf{x}^*$ with $V^* = V(\mathbf{x}^*)$. Correspondingly, using strong convexity of $V$, we can obtain the relation:

$$2\hat{H}(V(\mathbf{x}) - V^*) \leq \|\nabla V(\mathbf{x})\|^2 \text{ for all } \mathbf{x} \in \mathbb{R}^N. \tag{10}$$

From strong convexity and the Lipschitz continuous property of $\nabla f_j$, the constants $H_m$ and $\gamma_m$ further satisfy $H_m \leq \gamma_m$ and hence, $\hat{H} \leq \hat{\gamma}$.

Next, we introduce two key lemmas that will help establish our main theoretical guarantees. Due to space limitations, all proofs are deferred to the supplementary material in Section 7.

**Lemma 1.** *Under Assumptions 1 and 2, the iterates of CDSGD satisfy $\forall k \in \mathbb{N}$:*

$$\mathbb{E}[V(\mathbf{x}_{k+1})] - V(\mathbf{x}_k) \leq -\alpha \nabla V(\mathbf{x}_k)^T \mathbb{E}[\nabla \mathcal{J}^i(\mathbf{x}_k)] + \frac{\hat{\gamma}}{2} \alpha^2 \mathbb{E}[\|\nabla \mathcal{J}^i(\mathbf{x}_k)\|^2] \qquad (11)$$

At a high level, since $\mathbb{E}[\nabla \mathcal{J}^i(\mathbf{x}_k)]$ is the unbiased estimate of $\nabla V(\mathbf{x}_k)$, using the updates $\nabla \mathcal{J}^i(\mathbf{x}_k)$ will lead to sufficient decrease in the Lyapunov function. However, unbiasedness is not enough, and we also need to control higher order moments of $\nabla \mathcal{J}^i(\mathbf{x}_k)$ to ensure convergence. Specifically, we consider the variance of $\nabla \mathcal{J}^i(\mathbf{x}_k)$:

$$Var[\nabla \mathcal{J}^i(\mathbf{x}_k)] := \mathbb{E}[\|\nabla \mathcal{J}^i(\mathbf{x}_k)\|^2] - \|\mathbb{E}[\nabla \mathcal{J}^i(\mathbf{x}_k)]\|^2 \qquad (12)$$

To bound the variance of $\nabla \mathcal{J}^i(\mathbf{x}_k)$, we use a standard assumption presented in [25] in the context of (centralized) deep learning. Such an assumption aims at providing an upper bound for the "gradient noise" caused by the randomness in the minibatch selection at each iteration.

**Assumption 3.** *a) There exist scalars $\zeta_2 \geq \zeta_1 > 0$ such that $\nabla V(\mathbf{x}_k)^T \mathbb{E}[\nabla \mathcal{J}^i(\mathbf{x}_k)] \geq \zeta_1 \|\nabla V(\mathbf{x}_k)\|^2$ and $\|\mathbb{E}[\nabla \mathcal{J}^i(\mathbf{x}_k)]\| \leq \zeta_2 \|\nabla V(\mathbf{x}_k)\|$ for all $k \in \mathbb{N}$; b) There exist scalars $Q \geq 0$ and $Q_V \geq 0$ such that $Var[\nabla \mathcal{J}^i(\mathbf{x}_k)] \leq Q + Q_V \|\nabla V(\mathbf{x}_k)\|^2$ for all $k \in \mathbb{N}$.*

*Remark* 1. While Assumption 3(a) guarantees the sufficient descent of $V$ in the direction of $-\nabla \mathcal{J}^i(\mathbf{x}_k)$, Assumption 3(b) states that the variance of $\nabla \mathcal{J}^i(\mathbf{x}_k)$ is bounded above by the second moment of $\nabla V(\mathbf{x}_k)$. The constant $Q$ can be considered to represent the second moment of the "gradient noise" in $\nabla \mathcal{J}^i(\mathbf{x}_k)$. Therefore, the second moment of $\nabla \mathcal{J}^i(\mathbf{x}_k)$ can be bounded above as $\mathbb{E}[\|\nabla \mathcal{J}^i(\mathbf{x}_k)\|^2] \leq Q + Q_m \|\nabla V(\mathbf{x}_k)\|^2$, where $Q_m := Q_V + \zeta_2^2 \geq \zeta_1^2 > 0$.

**Lemma 2.** *Under Assumptions 1, 2, and 3, the iterates of CDSGD satisfy $\forall k \in \mathbb{N}$:*

$$\mathbb{E}[V(\mathbf{x}_{k+1})] - V(\mathbf{x}_k) \leq -(\zeta_1 - \frac{\hat{\gamma}}{2} \alpha Q_m) \alpha \|\nabla V(\mathbf{x}_k)\|^2 + \frac{\hat{\gamma}}{2} \alpha^2 Q . \qquad (13)$$

In Lemma 2, the first term is strictly negative if the step size satisfies the following necessary condition:

$$0 < \alpha \leq \frac{2\zeta_1}{\hat{\gamma} Q_m} \qquad (14)$$

However, in latter analysis, when such a condition is substituted into the convergence analysis, it may produce a larger upper bound. For obtaining a tight upper bound, we impose a sufficient condition for the rest of analysis as follows:

$$0 < \alpha \leq \frac{\zeta_1}{\hat{\gamma} Q_m} \qquad (15)$$

As $\hat{\gamma}$ is a function of $\alpha$, the above inequality can be rewritten as $0 < \alpha \leq \frac{\zeta_1 - (1 - \lambda_N(\Pi)) Q_m}{\gamma_m Q_m}$.

# 4 Main Results

We now present our main theoretical results establishing the convergence of CDSGD. First, we show that for most generic loss functions (whether convex or not), CDSGD achieves *consensus* across different agents in the graph, provided the step size (which is fixed across iterations) does not exceed a natural upper bound.

**Proposition 1.** *(Consensus with fixed step size) Under Assumptions 1 and 2, the iterates of CDSGD (Algorithm 1) satisfy $\forall k \in \mathbb{N}$:*

$$\mathbb{E}[\|x_k^j - s_k\|] \leq \frac{\alpha L}{1 - \lambda_2(\Pi)} \qquad (16)$$

*where $\alpha$ satisfies $0 < \alpha \leq \frac{\zeta_1 - (1 - \lambda_N(\Pi)) Q_m}{\gamma_m Q_m}$ and $L$ is an upper bound of $\mathbb{E}[\|\mathbf{g}(\mathbf{x}_k)\|]$, $\forall k \in \mathbb{N}$ (defined properly and discussed in Lemma 4 in the supplementary section 7) and $s_k = \frac{1}{N} \sum_{j=1}^{N} x_k^j$ represents the average parameter estimate.*

The proof of this proposition can be adapted from [26, Lemma 1].

Next, we show that for strongly convex loss functions, CDSGD converges linearly to a neighborhood of the global optimum.

**Theorem 1.** *(Convergence of CDSGD with fixed step size, strongly convex case) Under Assumptions 1, 2 and 3, the iterates of CDSGD satisfy the following inequality $\forall k \in \mathbb{N}$:*

$$\mathbb{E}[V(\mathbf{x}_k) - V^*] \leq (1 - \alpha \hat{H}\zeta_1)^{k-1}(V(\mathbf{x}_1) - V^*) + \frac{\alpha^2 \hat{\gamma} Q}{2} \sum_{l=0}^{k-1}(1 - \alpha \hat{H}\zeta_1)^l$$

$$= (1 - (\alpha H_m + 1 - \lambda_2(\Pi))\zeta_1)^{k-1}(V(\mathbf{x}_1) - V^*) \qquad (17)$$

$$+ \frac{(\alpha^2 \gamma_m + \alpha(1 - \lambda_N(\Pi)))Q}{2} \sum_{l=0}^{k-1}(1 - (\alpha H_m + 1 - \lambda_2(\Pi))\zeta_1)^l$$

*when the step size satisfies $0 < \alpha \leq \frac{\zeta_1 - (1 - \lambda_N(\Pi))Q_m}{\gamma_m Q_m}$.*

A detailed proof is presented in the supplementary section 7. We observe from Theorem 1 that the sequence of Lyapunov function values $\{V(\mathbf{x}_k)\}$ converges *linearly* to a *neighborhood* of the optimal value, i.e., $\lim_{k \to \infty} \mathbb{E}[V(\mathbf{x}_k) - V^*] \leq \frac{\alpha \hat{\gamma} Q}{2\hat{H}\zeta_1} = \frac{(\alpha \gamma_m + 1 - \lambda_N(\Pi))Q}{2(H_m + \alpha^{-1}(1 - \lambda_2(\Pi)))\zeta_1}$. We also observe that the term on the right hand side decreases with the spectral gap of the agent interaction matrix $\Pi$, i.e., $1 - \lambda_2(\Pi)$, which suggests an interesting relation between convergence and topology of the graph. Moreover, we observe that the upper bound is proportional to the step size parameter $\alpha$, and smaller step sizes lead to smaller radii of convergence. (However, choosing a very small step-size may negatively affect the convergence *rate* of the algorithm). Finally, if the gradient in this context is *not* stochastic (i.e., the parameter $Q = 0$), then linear convergence to the optimal value is achieved, which matches known rates of convergence with (centralized) gradient descent under strong convexity and smoothness assumptions.

*Remark* 2. Since $\mathbb{E}[\frac{N}{n}\mathbf{1}^T\mathbf{F}(\mathbf{x}_k)] \leq \mathbb{E}[V(\mathbf{x}_k)]$ and $\frac{N}{n}\mathbf{1}^T\mathbf{F}(\mathbf{x}^*) = V^*$, the sequence of objective function values are themselves upper bounded as follows: $\mathbb{E}[\frac{N}{n}\mathbf{1}^T\mathbf{F}(\mathbf{x}_k) - \frac{N}{n}\mathbf{1}^T\mathbf{F}(\mathbf{x}^*)] \leq \mathbb{E}[V(\mathbf{x}_k) - V^*]$. Therefore, using Theorem 1 we can establish analogous convergence rates in terms of the true objective function values $\{\frac{N}{n}\mathbf{1}^T\mathbf{F}(\mathbf{x}_k)\}$ as well.

The above convergence result for CDSGD is limited to the case when the objective functions are strongly convex. However, most practical deep learning systems (such as convolutional neural network learning) involve optimizing over highly *non-convex* objective functions, which are much harder to analyze. Nevertheless, we show that even under such situations, CDSGD exhibits a (weaker) notion of convergence.

**Theorem 2.** *(Convergence of CDSGD with fixed step size, nonconvex case) Under Assumptions 1, 2, and 3, the iterates of CDSGD satisfy $\forall m \in \mathbb{N}$:*

$$\mathbb{E}[\sum_{k=1}^{m} \|\nabla V(\mathbf{x}_k)\|^2] \leq \frac{\hat{\gamma} m \alpha Q}{\zeta_1} + \frac{2(V(\mathbf{x}_1) - V_{inf})}{\zeta_1 \alpha}$$

$$= \frac{(\gamma_m \alpha + 1 - \lambda_N(\Pi))mQ}{\zeta_1} + \frac{2(V(\mathbf{x}_1) - V_{inf})}{\zeta_1 \alpha}. \qquad (18)$$

*when the step size satisfies $0 < \alpha \leq \frac{\zeta_1 - (1 - \lambda_N(\Pi))Q_m}{\gamma_m Q_m}$.*

*Remark* 3. Theorem 2 states that when in the absence of "gradient noise" (i.e., when $Q = 0$), the quantity $\mathbb{E}[\sum_{k=1}^{m} \|\nabla V(\mathbf{x}_k)\|^2]$ remains finite. Therefore, necessarily $\{\|\nabla V(\mathbf{x}_k)\|\} \to 0$ and the estimates approach a stationary point. On the other hand, if the gradient calculations are stochastic, then a similar claim cannot be made. However, for this case we have the upper bound $\lim_{m \to \infty} \mathbb{E}[\frac{1}{m}\sum_{k=1}^{m} \|\nabla V(\mathbf{x}_k)\|^2] \leq \frac{(\gamma_m \alpha + 1 - \lambda_N(\Pi))Q}{\zeta_1}$. This tells us that while we cannot guarantee convergence in terms of sequence of objective function values, we can still assert that the average of the second moment of gradients is strictly bounded from above even for the case of nonconvex objective functions.

Moreover, the upper bound cannot be solely controlled via the step-size parameter $\alpha$ (which is different from what is implied in the strongly convex case by Theorem 1). In general, the upper bound becomes tighter as $\lambda_N(\Pi)$ increases; however, an increase in $\lambda_N(\Pi)$ may result in a commensurate increase in $\lambda_2(\Pi)$, leading to worse connectivity in the graph and adversely affecting consensus among agents. Again, our upper bounds are reflective of interesting tradeoffs between consensus and convergence in the gradients, and their dependence on graph topology.

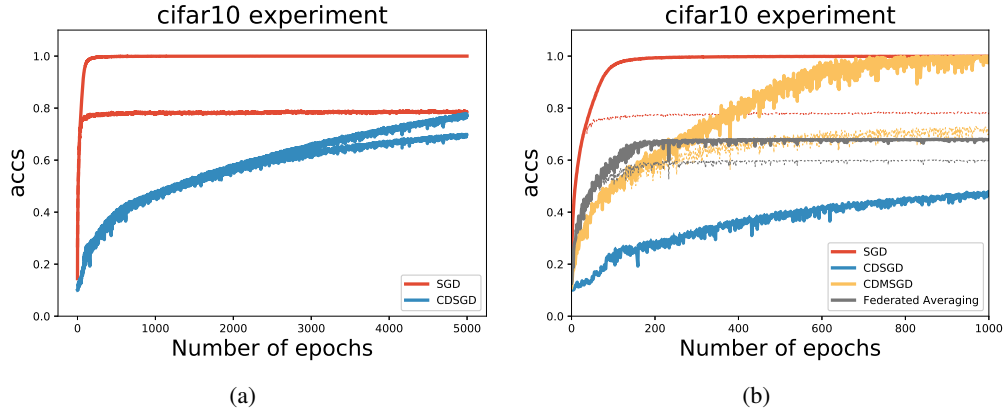

Figure 1: *Average training (solid lines) and validation (dash lines) accuracy for (a) comparison of CDSGD with centralized SGD and (b) CDMSGD with Federated average method*

The above results are for *fixed* step size $\alpha$, and we can prove complementary results for CDSGD even for the (more prevalent) case of *diminishing* step size $\alpha_k$. These are presented in the supplementary material due to space constraints.

## 5    Experimental Results

This section presents the experimental results using the benchmark image recognition dataset, CIFAR-10. We use a deep convolutional nerual network (CNN) model (with 2 convolutional layers with 32 filters each followed by a max pooling layer, then 2 more convolutional layers with 64 filters each followed by another max pooling layer and a dense layer with 512 units, ReLU activation is used in convolutional layers) to validate the proposed algorithm. We use a fully connected topology with 5 agents and uniform agent interaction matrix except mentioned otherwise. A mini-batch size of 128 and a fixed step size of 0.01 are used in these experiments. The experiments are performed using Keras and TensorFlow [27, 28] and the codes will be made publicly available soon. While we included the training and validation accuracy plots for the different case studies here, the corresponding training loss plots, results with other becnmark datasets such as MNIST and CIFAR-100 and decaying as well as different fixed step sizes are presented in the supplementary section 7.

### 5.1    Performance comparison with benchmark methods

We begin with comparing the accuracy of CDSGD with that of the centralized SGD algorithm as shown in Fig. 1(a). While the CDSGD convergence rate is significantly slower compared to SGD as expected, it is observed that CDSGD can eventually achieve high accuracy, comparable with centralized SGD. However, another interesting observation is that the generalization gap (the difference between training and validation accuracy as defined in [29]) for the proposed CDSGD algorithm is significantly smaller than that of SGD which is an useful property. We also compare both CDSGD and CDMSGD with the Federated averaging SGD (FedAvg) algorithm which also performs data parallelization (see Fig. 1(b)). For the sake of comparison, we use same number of agents and choose $E = 1$ and $C = 1$ as the hyperparameters in the FedAvg algorithm as it is close to a fully connected topology scenario as considered in the CDSGD and CDMSGD experiments. As CDSGD is significantly slow, we mainly compare the CDMSGD with FedAvg which have similar convergence rates (CDMSGD being slightly slower). The main observation is that CDMSGD performs better than FedAvg at the steady state and can achieve centralized SGD level performance. It is important to note that FedAvg does not perform decentralized computation. Essentially it runs a brute force parameter averaging on a central parameter server at every epoch (i.e., consensus at every epoch) and then broadcasts the updated parameters to the agents. Hence, it tends to be slightly faster than CDMSGD which uses a truly decentralized computation over a network.

### 5.2    Effect of network size and topology

In this section, we investigate the effects of network size and topology on the performance of the proposed algorithms. Figure 2(a) shows the change in training performance as the number of agents grow from 2 to 8 and to 16. Although with increase in number of agents, the convergence rate slows down, all networks are able to achieve similar accuracy levels. Finally, we investigate the impact of network sparsity (as quantified by the second largest eigenvalue) on the learning performance. The primary observation is convergence of average accuracy value happens faster for sparser networks

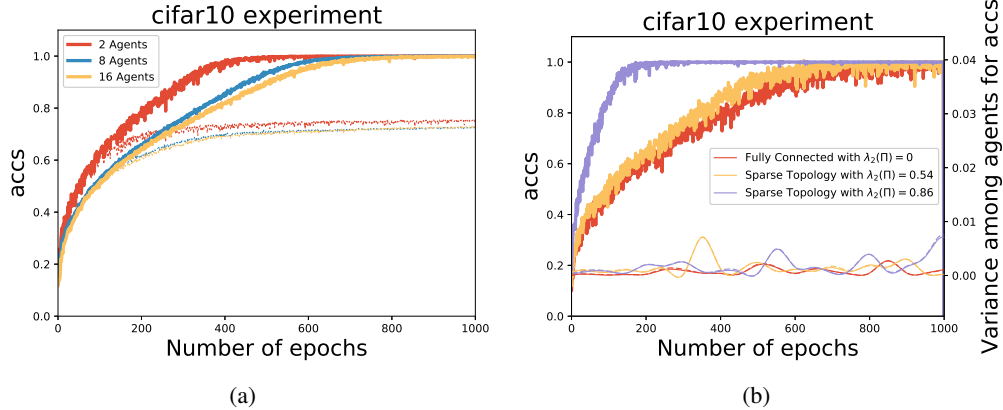

Figure 2: *Average training (solid lines) and validation (dash lines) accuracy along with accuracy variance over agents for CDMSGD algorithm with (a) varying network size and (b) varying network topology*

(higher second largest eigenvalue). This is similar to the trend observed for FedAvg algorithm while reducing the Client fraction ($C$) which makes the (stochastic) agent interaction matrix sparser. However, from the plot of the variance of accuracy values over agents (a smooth version using moving average filter), it can be observed that the level of consensus is more stable for denser networks compared to that for sparser networks. This is also expected as discussed in Proposition 1. Note, with the availability of a central parameter server (as in federated averaging), sparser topology may be useful for a faster convergence, however, consensus (hence, topology density) is critical for a collaborative learning paradigm with decentralized computation.

# 6   Conclusion and Future Work

This paper addresses the collaborative deep learning (and many other machine learning) problem in a completely distributed manner (i.e., with data parallelism and decentralized computation) over networks with fixed topology. We establish a consensus based distributed SGD framework and proposed associated learning algorithms that can prove to be extremely useful in practice. Using a Lyapunov function construction approach, we show that the proposed CDSGD algorithm can achieve linear convergence rate with sufficiently small fixed step size and sublinear convergence rate with diminishing step size (see supplementary section 7 for details) for strongly convex and Lipschitz differentiable objective functions. Moreover, decaying gradients can be observed for the nonconvex objective functions using CDSGD. Relevant experimental results using benchmark datasets show that CDSGD can achieve centralized SGD level accuracy with sufficient training epochs while maintaining a significantly low generalization error. The momentum variant of the proposed algorithm, CDMSGD can outperform recently proposed FedAvg algorithm which also uses data parallelism but does not perform a decentralized computation, i.e., uses a central parameter server. The effects of network size and topology are also explored experimentally which conforms to the analytical understandings. While current and future research is focusing on extensive testing and validation of the proposed framework especially for large networks, a few technical research directions include: (i) collaborative learning with extreme non-IID data; (ii) collaborative learning over directed time-varying graphs; and (iii) understanding the dependencies between learning rate and consensus.

## Acknowledgments

This paper is based upon research partially supported by the USDA-NIFA under Award no. 2017-67021-25965, the National Science Foundation under Grant No. CNS-1464279 and No. CCF-1566281. Any opinions, findings, and conclusions or recommendations expressed in this material are those of the authors and do not necessarily reflect the views of the funding agencies.

## Footnotes

[1]Note that in our formulation, we are assuming that every agent has the same local objective function while in general distributed optimization problems they can be different.

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
