[Supplementary Material]

# 7 Supplementary Materials for "Collaborative Deep Learning in Fixed Topology Networks"

## 7.1 Additional analytical results and proofs

We begin with proofs of the lemmas and theorems that are presented in the main body of the paper without proof. The statements of the lemmas and theorems are presented again for completeness.

**Lemma 1**: Let Assumptions 1 and 2 hold. The iterates of CDSGD (Algorithm 1) satisfy the following inequality $\forall k \in \mathbb{N}$:

$$\mathbb{E}[V(\mathbf{x}_{k+1})] - V(\mathbf{x}_k) \leq -\alpha_k \nabla V(\mathbf{x}_k)^T \mathbb{E}[\nabla \mathcal{J}^i(\mathbf{x}_k)] + \frac{\hat{\gamma}}{2}\alpha_k^2 \mathbb{E}[\|\nabla \mathcal{J}^i(\mathbf{x}_k)\|^2] \tag{19}$$

*Proof.* By Assumption 1, the iterates generated by CDSGD satisfy

$$\begin{aligned} V(\mathbf{x}_{k+1}) - V(\mathbf{x}_k) &\leq \nabla V(\mathbf{x}_k)^T(\mathbf{x}_{k+1} - \mathbf{x}_k) + \frac{1}{2}\hat{\gamma}\|\mathbf{x}_{k+1} - \mathbf{x}_k\|^2 \\ &= -\alpha \nabla V(\mathbf{x}_k)^T \nabla \mathcal{J}^i(\mathbf{x}_k) + \frac{1}{2}\hat{\gamma}\alpha^2 \|\nabla \mathcal{J}^i(\mathbf{x}_k)\|^2 \end{aligned} \tag{20}$$

Taking expectations on both sides, we can obtain

$$\mathbb{E}[V(\mathbf{x}_{k+1}) - V(\mathbf{x}_k)] \leq \mathbb{E}[-\alpha \nabla V(\mathbf{x}_k)^T \nabla \mathcal{J}^i(\mathbf{x}_k) + \frac{1}{2}\hat{\gamma}\alpha^2 \|\nabla \mathcal{J}^i(\mathbf{x}_k)\|^2] \tag{21}$$

While $V(\mathbf{x}_k)$ is deterministic, $V(\mathbf{x}_{k+1})$ is stochastic due to the random sampling aspect. Therefore, we have

$$\mathbb{E}[V(\mathbf{x}_{k+1})] - V(\mathbf{x}_k) \leq -\alpha \nabla V(\mathbf{x}_k)^T \mathbb{E}[\nabla \mathcal{J}^i(\mathbf{x}_k)] + \frac{1}{2}\hat{\gamma}\alpha^2 \mathbb{E}[\|\nabla \mathcal{J}^i(\mathbf{x}_k)\|^2] \tag{22}$$

which completes the proof. $\qquad\square$

**Lemma 2**: Let Assumptions 1, 2, and 3 hold. The iterates of CDSGD (Algorithm 1) satisfy the following inequality $\forall k \in \mathbb{N}$:

$$\mathbb{E}[V(\mathbf{x}_{k+1})] - V(\mathbf{x}_k) \leq -(\zeta_1 - \frac{\hat{\gamma}}{2}\alpha Q_m)\alpha\|\nabla V(\mathbf{x}_k)\|^2 + \frac{\hat{\gamma}}{2}\alpha^2 Q \tag{23}$$

*Proof.* Recalling Lemma 1 and using Assumption 3 and Remark 1, we have

$$\begin{aligned} \mathbb{E}[V(\mathbf{x}_{k+1})] - V(\mathbf{x}_k) &\leq -\zeta_1\alpha\|\nabla V(\mathbf{x}_k)\|^2 + \frac{\hat{\gamma}}{2}\alpha^2 \mathbb{E}[\|\nabla \mathcal{J}^i(\mathbf{x}_k)\|^2] \\ &\leq -\zeta_1\alpha\|\nabla V(\mathbf{x}_k)\|^2 + \frac{\hat{\gamma}}{2}\alpha^2(Q + Q_m\|\nabla V(\mathbf{x}_k)\|^2) \\ &= -(\zeta_1 - \frac{\hat{\gamma}}{2}\alpha Q_m)\alpha\|\nabla V(\mathbf{x}_k)\|^2 + \frac{\hat{\gamma}}{2}\alpha^2 Q \end{aligned} \tag{24}$$

which completes the proof. $\qquad\square$

In order to prove Propositon 1, several auxiliary technical lemmas are presented first.

**Lemma 3.** *V has a lower bound denoted by $V_{inf}$ over an open set which contains the iterates $\{\mathbf{x}_k\}$ generated by CDSGD (Algorithm 1).*

Lemma 3 can be obtained as each $f_j$ is proper and coercive. Such a lemma is able to help characterize the nonconvex case in which the global optimum may not be achieved.

**Lemma 4.** *Let Assumption 1 holds. There exists some constant $0 < L < \infty$ such that $\mathbb{E}[\|\mathbf{g}(\mathbf{x}_k)\|]] \le L$.*

The proof of Lemma 4 directly follows from the Assumption 1 c) and $L = \max_j L_j$.

**Theorem 1**(Convergence of CDSGD with fixed step size, strongly convex case): Let Assumptions 1, 2 and 3 hold. The iterates of CDSGD (Algorithm 1) satisfy the following inequality $\forall k \in \mathbb{N}$, when the step size satisfies

$$0 < \alpha \le \frac{\zeta_1 - (1 - \lambda_N(\Pi))Q_m}{\gamma_m Q_m}$$

$$
\begin{aligned}
\mathbb{E}[V(\mathbf{x}_k) - V^*] &\le (1 - \alpha\hat{H}\zeta_1)^{k-1}(V(\mathbf{x}_1) - V^*) + \frac{\alpha^2\hat{\gamma}Q}{2}\sum_{l=0}^{k-1}(1 - \alpha\hat{H}\zeta_1)^l \\
&= (1 - (\alpha H_m + 1 - \lambda_2(\Pi))\zeta_1)^{k-1}(V(\mathbf{x}_1) - V^*) \\
&\quad + \frac{(\alpha^2\gamma_m + \alpha(1 - \lambda_N(\Pi)))Q}{2}\sum_{l=0}^{k-1}(1 - (\alpha H_m + 1 - \lambda_2(\Pi))\zeta_1)^l
\end{aligned}
\tag{25}
$$

*Proof.* Recalling Lemma 2 and using Eq. 10 yield that

$$
\begin{aligned}
\mathbb{E}[V(\mathbf{x}_{k+1})] - V(\mathbf{x}_k) &\le -(\zeta_1 - \frac{\hat{\gamma}}{2}\alpha Q_m)\alpha\|\nabla V(\mathbf{x}_k)\|^2 + \frac{\hat{\gamma}}{2}\alpha^2 Q \\
&\le -\frac{1}{2}\alpha\zeta_1\|\nabla(\mathbf{x}_k)\|^2 + \frac{\alpha^2\hat{\gamma}Q}{2} \\
&\le -\alpha\zeta_1\hat{H}(V(\mathbf{x}_k) - V^*) + \frac{\alpha^2\hat{\gamma}Q}{2}
\end{aligned}
\tag{26}
$$

The second inequality follows from the relation: $\alpha \le \frac{\zeta_1}{\hat{\gamma}Q_m}$, which is implied by: $\alpha \le \frac{\zeta_1 - (1 - \lambda_N(\Pi))Q_m}{\gamma_m Q_m}$. The third inequality follows from the strong convexity. The expectation taken in the above inequalities is only related to $\mathbf{x}_{k+1}$. Hence, recursively taking the expectation and subtracting $V^*$ from both sides requires the following inequality to hold

$$\mathbb{E}[V(\mathbf{x}_{k+1}) - V^*] \le (1 - \alpha\hat{H}\zeta_1)\mathbb{E}[V(\mathbf{x}_k) - V^*] + \frac{\alpha^2\hat{\gamma}Q}{2} \tag{27}$$

As $0 < \alpha\hat{H}\zeta_1 \le \frac{\hat{H}\zeta_1^2}{\hat{\gamma}Q_m} \le \frac{\hat{H}\zeta_1^2}{\hat{\gamma}\zeta_1^2} = \frac{\hat{H}}{\hat{\gamma}} \le 1$, the conclusion follows by applying Eq. 27 recursively through iteration $k \in \mathbb{N}$. $\square$

**Theorem 2**(Convergence of CDSGD with fixed step size, nonconvex case): Let Assumptions 1, 2, and 3 hold. The iterates of CDSGD (Algorithm 1) satisfy the following inequality $\forall m \in \mathbb{N}$, when the step size satisfies

$$0 < \alpha \le \frac{\zeta_1 - (1 - \lambda_N(\Pi))Q_m}{\gamma_m Q_m}$$

$$
\begin{aligned}
\mathbb{E}[\sum_{k=1}^{m}\|\nabla V(\mathbf{x}_k)\|^2] &\le \frac{\hat{\gamma}m\alpha Q}{\zeta_1} + \frac{2(V(\mathbf{x}_1) - V_{\inf})}{\zeta_1\alpha} \\
&= \frac{(\gamma_m\alpha + 1 - \lambda_N(\Pi))mQ}{\zeta_1} + \frac{2(V(\mathbf{x}_1) - V_{\inf})}{\zeta_1\alpha}
\end{aligned}
\tag{28}
$$

*Proof.* Recalling Lemma 2, and also taking the expectation lead to the following relation,

$$\mathbb{E}[V(\mathbf{x}_{k+1})] - \mathbb{E}[V(\mathbf{x}_k)] \le -(\zeta_1 - \frac{\hat{\gamma}\alpha Q_m}{2})\alpha\mathbb{E}[\|\nabla V(\mathbf{x}_k)\|^2] + \frac{\hat{\gamma}\alpha^2 Q}{2} \tag{29}$$

As the step size satisfies that $\alpha \le \frac{\zeta_1}{\hat{\gamma}Q_m}$, it results in

$$\mathbb{E}[V(\mathbf{x}_{k+1})] - \mathbb{E}[V(\mathbf{x}_k)] \le -\frac{\zeta_1\alpha}{2}\mathbb{E}[\|\nabla V(\mathbf{x}_k)\|^2] + \frac{\alpha^2\hat{\gamma}Q}{2} \tag{30}$$

Applying the above inequality from 1 to $m$ and summing them up can give the following relation

$$V_{\inf} - V(\mathbf{x}_1) \leq \mathbb{E}[V(\mathbf{x}_{k+1})] - V(\mathbf{x}_1) \leq -\frac{\zeta_1 \alpha}{2} \sum_{k=1}^{m} \mathbb{E}[\|\nabla V(\mathbf{x}_k)\|^2] + \frac{m\alpha^2 \hat{\gamma} Q}{2} \qquad (31)$$

The last inequality follows from the Lemma 3. Rearrangement of the above inequality and substituting $\hat{\gamma} = \gamma_m + \alpha^{-1}(1 - \lambda_N(\Pi))$ into it yield the desired result. □

## 7.2 Proof with Diminishing Step Size

From results presented in section 4, it can be concluded that when the step size is fixed, the function value can only converge near the optimal value. However, in many deep learning models, noisy gradient is quite common due to the random data sampling. Hence, such a situation requires the step size to be adaptive and then with noise, the function value sequence is able to converge to the optimal value. Let $\{\alpha_k\}$ be defined as a diminishing step size sequence that satisfies the following properties:

$$\alpha_k > 0, \ \sum_{k=0}^{\infty} \alpha_k = \infty, \ \sum_{k=0}^{\infty} \alpha_k^2 < \infty$$

The implication of the above properties is that $\lim_{k\to\infty} \alpha_k = 0$. The next proposition states that when the step size is diminishing, consensus can be achieved asymptotically, i.e., $\lim_{k\to\infty} \mathbb{E}[\|x_k^j - s_k\|] = 0$.

**Proposition 2.** *(Consensus with diminishing step size) Let Assumptions 1 and 2 hold. The iterates of CDSGD (Algorithm 1) satisfy the following inequality $\forall k \in \mathbb{N}$, when $\alpha_k$ is diminishing,*

$$\lim_{k\to\infty} \mathbb{E}[\|x_k^j - s_k\|] = 0 \qquad (32)$$

The proof is adapted from the Lemma 1 in [26], Lemmas 5 and 6 in [30].

Recalling the algorithm CDSGD

$$\mathbf{x}_{k+1} = \mathbf{w}_k - \alpha_k \mathbf{g}(\mathbf{x}_k) = \mathbf{x}_k - \alpha_k(\mathbf{g}(\mathbf{x}_k) + \frac{1}{\alpha_k}(\mathbf{x}_k - \mathbf{w}_k))$$

We define $\nabla \hat{\mathcal{J}}^i(\mathbf{x}_k) = \nabla \mathcal{J}^i(\mathbf{x}_k, \alpha_k) = \mathbf{g}(\mathbf{x}_k) + \frac{1}{\alpha_k}(\mathbf{x}_k - \mathbf{w}_k)$, and the following Lyapunov function

$$\hat{V}(\mathbf{x}) = V(\mathbf{x}, \alpha_k) := \frac{N}{n} \mathbf{1}^T \mathbf{F}(\mathbf{x}) + \frac{1}{2\alpha_k} \|\mathbf{x}\|_{I-\Pi}^2 \qquad (33)$$

The general Lyapunov function is a function of the diminishing step size $\alpha_k$. However, the step size is independent of the variable $\mathbf{x}$ such that it only affects the magnitude of $\|\mathbf{x}\|_{I-\Pi}^2$ along with iterations. Note, from Proposition 2, we have that each agent eventually reaches the consensus with diminishing step size. Hence, the term $\frac{1}{2\alpha_k}\|\mathbf{x}\|_{I-\Pi}^2$ should not increase with increase in $k$ as the step size $\alpha_k \to 0$ for $k \to \infty$. To show that CDSGD with diminishing step size enables convergence to the optimal value, the necessary lemmas and assumptions are directly used from the previous part of the paper with modified constants.

We next show that the Lyapunov function and stochastic Lyapunov gradient with the diminishing step size are bounded. More formally, we aim to show that $\|\nabla \hat{\mathcal{J}}^i(\mathbf{x}_k)\|$ is bounded above for all $k \in \mathbb{N}$. We have, $\|\nabla \hat{\mathcal{J}}^i(\mathbf{x}_k)\| \leq \|\mathbf{g}(\mathbf{x}_k)\| + \frac{1}{\alpha_k}\|(I - \Pi)\mathbf{x}_k\|$ and $\mathbf{g}(\mathbf{x}_k)$ is bounded. Therefore, we have to show that $\frac{1}{\alpha_k}\|(I - \Pi)\mathbf{x}_k\|$ is bounded for all $k \in \mathbb{N}$.

**Lemma 5.** *Let Assumptions 1 and 2 hold. The iterates of CDSGD (Algorithm 1) satisfy the following inequality $\forall k \in \mathbb{N}$, when the step size is diminishing and satisfies that*

$$0 < \alpha_0 \leq \frac{\hat{\zeta}_1 - (1 - \lambda_N(\Pi))\hat{Q}_m}{\gamma_m \hat{Q}_m},$$

$$\frac{1}{\alpha_k} \mathbb{E}[\|(I - \Pi)\mathbf{x}_k\|] < \infty \qquad (34)$$

*and*

$$\lim_{k\to\infty} \mathbb{E}[\|(I - \Pi)\mathbf{x}_k\|] = 0. \qquad (35)$$

$\hat{\zeta}_1, \hat{Q}_m$ *correspond to* $\hat{V}$.

The proof of Lemma 5 requires another auxiliary technical lemma as follows.

**Lemma 6.** *Let Assumptions 1 and 2 hold. The iterates of CDSGD (Algorithm 1) satisfy the following inequality $\forall k \in \mathbb{N}$, when the step size is diminishing and satisfies that*

$$0 < \alpha_0 \le \frac{\hat{\zeta}_1 - (1 - \lambda_N(\Pi))\hat{Q}_m}{\gamma_m \hat{Q}_m},$$

$$\sum_{k=2}^{\infty} \alpha_k \mathbb{E}[\|(I - \Pi)\mathbf{x}_k\|] < \infty. \tag{36}$$

*Proof.* Recalling the CDSGD algorithm,

$$\mathbf{x}_{k+1} = \Pi\mathbf{x}_k - \alpha_k \mathbf{g}(\mathbf{x}_k) \tag{37}$$

Applying the above equality from 1 to $k-1$ yields that

$$\mathbf{x}_k = \Pi^{k-1}\mathbf{x}_1 - \sum_{l=1}^{k-1} \alpha_l \Pi^{k-1-l}\mathbf{g}(\mathbf{x}_l) \tag{38}$$

Setting $\mathbf{x}_1 = 0$ results in that $\mathbf{x}_k = -\sum_{l=1}^{k-1} \alpha_l \Pi^{k-1-l}\mathbf{g}(\mathbf{x}_l)$. With this setup, we have

$$\begin{aligned}
\sum_{k=2}^{\infty} \alpha_k \|(I - \Pi)\mathbf{x}_k\| &\le \sum_{k=2}^{\infty} \alpha_k \|I - \Pi\|\|\mathbf{x}_k\| \\
&\le \sum_{k=2}^{\infty} \alpha_k \|\sum_{l=1}^{k-1} \alpha_l \Pi^{k-1-l}\mathbf{g}(\mathbf{x}_l)\| \\
&\le \sum_{k=2}^{\infty} \alpha_k \sum_{l=1}^{k-1} \alpha_l \|\Pi^{k-1-l}\mathbf{g}(\mathbf{x}_l)\| \\
&\le \sum_{k=2}^{\infty} \alpha_k \sum_{l=1}^{k-1} \alpha_l \|\Pi^{k-1-l}\|\|\mathbf{g}(\mathbf{x}_l)\| \\
&\le \sum_{k=2}^{\infty} \alpha_k \sum_{l=1}^{k-1} \alpha_l \|\Pi\|^{k-1-l}\|\mathbf{g}(\mathbf{x}_l)\|
\end{aligned} \tag{39}$$

With the step size being nonincreasing, taking expectation on both side leads to

$$\mathbb{E}[\sum_{k=2}^{\infty} \alpha_k \|(I - \Pi)\mathbf{x}_k\|] \le \sum_{k=2}^{\infty} \sum_{l=1}^{k-1} \alpha_l^2 \|\Pi\|^{k-1-l}\mathbb{E}[\|\mathbf{g}(\mathbf{x}_l)\|] \tag{40}$$

As we discussed earlier, we consider that there exists a constant $L$ that bounds $\mathbb{E}[\|\mathbf{g}(\mathbf{x}_k)\|]$ from above for $k \in \mathbb{N}$. Thus, the following relation can be obtained

$$\begin{aligned}
\mathbb{E}[\sum_{k=2}^{\infty} \alpha_k \|(I - \Pi)\mathbf{x}_k\|] &\le \sum_{k=2}^{\infty} \sum_{l=1}^{k-1} \alpha_l^2 \lambda_2(\Pi)^{k-1-l}\mathbb{E}[\|\mathbf{g}(\mathbf{x}_l)\|] \\
&\le L \sum_{k=2}^{\infty} \sum_{l=1}^{k-1} \alpha_l^2 \lambda_2(\Pi)^{k-1-l}
\end{aligned} \tag{41}$$

As $\sum_{k=1}^{\infty} \alpha_k^2 < \infty$ and $\lambda_2(\Pi) < 1$ then by Lemma 5 in [30], the desired result follows. □

*Proof of Lemma 5.* We first define that $h_k = \frac{\|(I-\Pi)\mathbf{x}_k\|}{\alpha_k}$. Hence, the result of Lemma 6 can be rewritten as $\sum_{k=2}^{\infty} \alpha_k^2 \mathbb{E}[h_k] < \infty$. By defining $h_m = \sup\{\mathbb{E}[h_k]\}$, we have $h_m \sum_{k=2}^{\infty} \alpha_k^2 < \infty$, which implies that $h_m < \infty$ as $\sum_{k=1}^{\infty} \alpha_k^2 < \infty$. Hence, it is immediately seen that $\mathbb{E}[h_k] < \infty$. As $k \to \infty, \alpha_k \to 0$ and $\frac{1}{\alpha_k}\mathbb{E}[\|(I-\Pi)\mathbf{x}_k\|] < \infty$, then $\lim_{k\to\infty} \mathbb{E}[\|(I-\Pi)\mathbf{x}_k\|] = 0$, which completes the proof. □

The implication of Lemma 5 is two folds: One can observe that $\hat{V}(\mathbf{x}_k)$ and $\nabla\hat{\mathcal{J}}^i(\mathbf{x}_k)$ are finite even with diminishing step size such that based on Definition 2 there exists a finite positive constant $\gamma'$ to allow the smoothness of $\hat{V}(\mathbf{x}_k)$ for all $k \in \mathbb{N}$ to hold true; another observation is that Assumption 3 still can be used in the main results. It can also be concluded that $\hat{V}$ is strongly convex with some constant $0 \leq H' < \infty$, where $H'$ corresponds to $\hat{V}$.

**Theorem 3.** *(Convergence of CDSGD with diminishing step size, strongly convex case) Let Assumptions 1, 2 and 3 hold. The iterates of CDSGD (Algorithm 1) satisfy the following inequality $\forall k \in \mathbb{N}$, when the step size is diminishing and satisfies that*

$$0 < \alpha_0 \leq \frac{\hat{\zeta}_1 - (1 - \lambda_N(\Pi))\hat{Q}_m}{\gamma_m \hat{Q}_m},$$

$$\mathbb{E}[\hat{V}(\mathbf{x}_k) - \hat{V}^*] \leq \beta^{k-2}(\hat{V}(\mathbf{x}_1) - \hat{V}^*) + \frac{\gamma'\hat{Q}}{2}\sum_{p=1}^{k-2}\beta^{k-p-2}\alpha_p^2$$

$$+ \frac{\gamma'\hat{Q}\alpha_{k-1}^2}{2} \tag{42}$$

*where $\sup\{1 - \alpha_k H'\hat{\zeta}_1\} \leq \beta < 1$, and $\gamma', \hat{Q}$ correspond to $\hat{V}$.*

*Proof.* As $\alpha_1 \leq \frac{\hat{\zeta}_1}{\gamma'\hat{Q}_m}$, then it can be obtained that $\alpha_k\gamma'\hat{Q}_m \leq \alpha_1\gamma'\hat{Q}_m \leq \hat{\zeta}_1$ for all $k \in \mathbb{N}$. Recalling Lemma 2 and Eq. 10, subtracting $\hat{V}^*$ from both sides, and taking the expectation yield the following relation

$$\mathbb{E}[\hat{V}(\mathbf{x}_{k+1}) - \hat{V}^*] \leq (1 - \alpha_k H'\hat{\zeta}_1)\mathbb{E}[\hat{V}(\mathbf{x}_k) - \hat{V}^*] + \frac{\gamma'\hat{Q}\alpha_k^2}{2} \tag{43}$$

Applying the above inequality recursively can give the following relation

$$\mathbb{E}[\hat{V}(\mathbf{x}_{k+1}) - \hat{V}^*] \leq (1 - \alpha_k H'\hat{\zeta}_1)(1 - \alpha_{k-1}H'\hat{\zeta}_1)\mathbb{E}[\hat{V}(\mathbf{x}_{k-1}) - \hat{V}^*]$$

$$+ (1 - \alpha_{k-1}H'\hat{\zeta}_1)\frac{\gamma'\hat{Q}\alpha_{k-1}^2}{2} + \frac{\gamma'\hat{Q}\alpha_k^2}{2} \tag{44}$$

By induction, the following can be obtained

$$\mathbb{E}[\hat{V}(\mathbf{x}_{k+1}) - \hat{V}^*] \leq$$

$$\prod_{q=1}^{k}(1 - \alpha_q H'\hat{\zeta}_1)(\hat{V}(\mathbf{x}_1) - \hat{V}^*) + \frac{\gamma'\hat{Q}}{2}\sum_{p=1}^{k-1}\prod_{r=p+1}^{k}(1 - \alpha_r H'\hat{\zeta}_1)\alpha_p^2$$

$$+ \frac{\gamma'\hat{Q}\alpha_k^2}{2} \tag{45}$$

As $H' \leq \gamma'$, it can be derived that $0 < \alpha_k H'\hat{\zeta}_1 \leq 1$ for all $k \in \mathbb{N}$. Therefore, $1 - \alpha_k H'\hat{\zeta}_1 \in [0, 1)$ such that we can define a positive constant $\beta$ satisfies that $\sup\{1 - \alpha_k H'\hat{\zeta}_1\} \leq \beta < 1$. Hence, combining the last inequalities together, we have

$$\mathbb{E}[\hat{V}(\mathbf{x}_{k+1}) - \hat{V}^*] \leq \beta^{k-1}(\hat{V}(\mathbf{x}_1) - \hat{V}^*) + \frac{\gamma'\hat{Q}}{2}\sum_{p=1}^{k-1}\beta^{k-p-1}\alpha_p^2$$

$$+ \frac{\gamma'\hat{Q}\alpha_k^2}{2} \tag{46}$$

which completes the proof by replacing $k + 1$ with $k$. $\square$

*Remark* 4. From Theorem 3, we can conclude that the function value sequence $\{\hat{V}(\mathbf{x}_k)\}$ asymptotically converges to the optimal value. (This holds regardless of whether the "gradient noise" parameter $\hat{Q}$ is zero or not.) In fact, we can establish the rate of convergence as follows: the first term on the right hand side decreases exponentially if $\beta. < 1$, and the last term decreases as quickly as $\alpha_k^2$. For

the middle term, we can use Lemma 3.1 of [31] that establishes bounds on the convolution of two scalar sequences. If we choose $t > 0$ such that $\alpha_k = \frac{1}{k^\epsilon + t}$, where $\epsilon \in (0.5, 1]$, then the necessary growth conditions on $\alpha_k$ are satisfied; substituting this into Theorem 3 yields the stated convergence rate of $\mathcal{O}(\frac{1}{k^\epsilon})$. In practice, $\alpha_k$ can be made adaptive to $\frac{\Theta}{k^\epsilon + t}$ for any constant $\Theta > 0$.

Similarly, we also present the convergence results for the nonconvex objective functions.

**Theorem 4.** *(Convergence of CDSGD with diminishing step size, nonconvex case) Let Assumptions 1, 2 and 3 hold. The iterates of CDSGD (Algorithm 1) satisfy the following inequality $\forall m \in \mathbb{N}$, when the step size is diminishing and satisfies that*

$$0 < \alpha_0 \leq \frac{\hat{\zeta}_1 - (1 - \lambda_N(\Pi))\hat{Q}_m}{\gamma_m \hat{Q}_m},$$

$$\mathbb{E}[\sum_{k=1}^m \alpha_k \|\nabla \hat{V}(\mathbf{x}_k)\|^2] \leq \frac{2(\hat{V}(\mathbf{x}_1) - \hat{V}_{inf})}{\hat{\zeta}_1} + \frac{\gamma' \hat{Q}}{\hat{\zeta}_1} \sum_{k=1}^m \alpha_k^2 \qquad (47)$$

*Proof.* Assume that $\alpha_k \gamma' \hat{Q}_m \leq \hat{\zeta}_1$ for all $k \in \mathbb{N}$. Based on Eq. 29 we consider the diminishing step size and Lyapunov function, then the following relation can be obtained

$$\mathbb{E}[\hat{V}(\mathbf{x}_{k+1})] - \mathbb{E}[\hat{V}(\mathbf{x}_k)] \leq -(\hat{\zeta}_1 - \frac{\gamma' \alpha_k \hat{Q}_m}{2})\alpha_k \mathbb{E}[\|\nabla \hat{V}(\mathbf{x}_k)\|^2] + \frac{\gamma' \alpha_k \hat{Q}}{2} \qquad (48)$$

Combining the condition for the step size yields the following inequality

$$\mathbb{E}[\hat{V}(\mathbf{x}_{k+1})] - \mathbb{E}[\hat{V}(\mathbf{x}_k)] \leq -\frac{\hat{\zeta}_1 \alpha_k}{2}\mathbb{E}[\|\nabla \hat{V}(\mathbf{x}_k)\|^2] + \frac{\gamma' \alpha_k \hat{Q}}{2} \qquad (49)$$

Applying the last inequality from 1 to $m$ and summing them up,

$$\hat{V}_{inf} - \mathbb{E}[\hat{V}(\mathbf{x}_1)] \leq \mathbb{E}[\hat{V}(\mathbf{x}_{k+1})] - \mathbb{E}[\hat{V}(\mathbf{x}_1)] \leq$$
$$-\frac{\hat{\zeta}_1}{2} \sum_{k=1}^m \alpha_k \mathbb{E}[\|\nabla \hat{V}(\mathbf{x}_k)\|^2] + \frac{\gamma' \hat{Q}}{2} \sum_{k=1}^m \alpha_k^2 \qquad (50)$$

Dividing by $\hat{\zeta}_1 / 2$ and rearranging the terms lead to the desired results. □

*Remark* 5. Compared to Theorem 2, Theorem 4 has shown the decaying of gradient $\|\nabla \hat{V}(\mathbf{x}_k)\|$ even with noise when the step size is diminishing in the nonconvex case. This is because when $k \to \infty$, the right hand side of Eq. 47 remains finite such that $\|\nabla \hat{V}(\mathbf{x}_k)\|^2$ approaches 0.

## 7.3 Additional pseudo-codes of the algorithms

Momentum methods have been regarded as effective methods to speed up the convergence in numerous optimization problems. While the Nesterov Momentum method has been extended widely to generate variants with provable global convergence properties, the global convergence analysis of Polyak Momentum methods is still quite challenging and an active research topic. Pseudo-codes

Figure 3: *Average training (solid lines) and validation (dash lines) loss for (a) CDSGD algorithm with SGD algorithm and (b) CDMSGD with Federated averaging method*

of CDSGD combined with Polyak momentum and Nesterov momentum methods are presented below.

---

**Algorithm 2:** CDSGD with Polyak Momentum

---

**Input** : $m$, $\alpha$, $N$, $\mu$ (momentum term)
**Initialize**: $x_0^j$, $v_0^j$
Distribute the training data set to $N$ agents
For each agent:
**for** $k = 0 : m$ **do**
    Randomly shuffle the corresponding data subset;
    $w_{k+1}^j = \sum_{l \in Nb(j)} \pi_{jl} x_k^l$
    $v_{k+1}^j = \mu v_k^j - \alpha_k g_j(x_k^j)$
    $x_{k+1}^j = w_{k+1}^j + v_{k+1}^j$
**end**

---

---

**Algorithm 3:** CDSGD with Nesterov Momentum

---

**Input** : $m$, $\alpha$, $N$, $\mu$
**Initialize**: $x_0^j$, $v_0^j$
Distribute the training data set to $N$ agents
For each agent:
**for** $k = 0 : m$ **do**
    Randomly shuffle the corresponding data subset
    $w_{k+1}^j = \sum_{l \in Nb(j)} \pi_{jl} x_k^l$
    $v_{k+1}^j = \mu v_k^j - \alpha_k g_j(x_k^j + \mu v_k^j)$
    $x_{k+1}^j = w_{k+1}^j + v_{k+1}^j$
**end**

---

## 7.4 Additional Experimental Results

We begin with a discussion on the training loss profiles for the CIFAR-10 results presented in the main body of the paper.

### 7.4.1 Comparison of the loss for benchmark methods

Figure 3 (a) shows the loss (in log scale) with respect to the number of epochs for SGD and CDSGD algorithms. The solid curve means training and the dash curve indicates validation. From the loss

Figure 4: *Average training (solid lines) and validation (dash lines) (a) loss and (b) accuracy for SGD, CDSGD, CDMSGD and Federated averaging method for the CIFAR-100 dataset (c) loss and (d) accuracy for SGD, CDSGD, CDMSGD and Federated averaging method for the MNIST dataset*

results, it can be observed that SGD has the sublinear convergence rate for training and dominates among the two methods during the training process. While for the validation, SGD performs poorly after around 70 epochs. However, CDSGD shows linear convergence rate (in log scale as discussed in the analysis) for both training and validation. Though, it takes a lot of more time compared to SGD for convergence, it eventually performs better than SGD in the validation data and the gap between the training and validation loss (i.e., the generalization gap [29]) is very less compared to that in SGD.

### 7.4.2 Results on CIFAR-100 dataset

For the experiments on the CIFAR-100 dataset, we use a CNN similar to that used for the CIFAR-10 dataset. While the results of CIFAR-100 also converges fast for SGD, CDMSGD and Federated Averaging SGD (FedAvg) algorithms (CDMSGD being the slowest) as shown in Figure 4, it can be seen that eventually, the loss converges better than the FedAvg algorithm. Similar to the observation made for the CIFAR-10 dataset, we observe that CDMSGD achieves significantly higher validation accuracy compared to FedAvg while approaching similar accuracy level as that of (centralized) SGD. It can also be seen that as expected CDSGD's convergence is very slow compared to the others.

### 7.4.3 Results on MNIST dataset

For the experiments on the MNIST dataset, the model used for training is a Deep Neural Network with 20 Fully Connected layers consisting of 50 ReLU units each and the output layer with 10 units having softmax activation. The model was trained using the catagorical cross-entropy loss. Figure 4(c & d) shows the loss and accuracy obtained over the number of epochs. In this case, while the accuracy levels are significantly higher as expected for the MNIST dataset, the trends remain

Figure 5: *Average training (solid lines) and validation (dash lines) (a) loss and (b) accuracy for SGD, MSGD and CDMSGD method for the MNIST dataset for decaying step size. (c) loss and (d) accuracy for CDMSGD for the MNIST data with different learning rates*

consistent with the results obtained for the other benchmark datasets of CIFAR-10 and CIFAR-100. Note, the generalization gap between the training and validation data for all the methods are very less (least for CDMSGD).

### 7.4.4 Effect of the decaying step size

Based on the analysis presented in section 7.2, it is evident that decaying step size has a significant effect on the accuracy as well as convergence. A performance comparison of SGD, Momentum SGD (MSGD) and CDMSGD with a decaying stepsize is performed using the MNIST dataset. It can be seen that the performance of the CDMSGD with decaying step size becomes slightly better than SGD with decaying step size while (centralized) MSGD has the best performance. Although CDMSGD sometimes suffers from large fluctuations, it demonstrates the least generalization gap among all the algorithms.

### 7.4.5 Effect of step size

The analysis presented in this paper shows that choice of step size is critical in terms of convergence as well as accuracy. To explore this aspect experimentally, we compare the performance of CDMSGD for three different fixed step sizes using MNIST data. The results are presented in 5 (c) & (d), where the (fixed) step size was varied from $0.1(1E-1)$ to $0.01(1E-2)$ and then to $0.001(1E-3)$. While the fastest convergence of the algorithm is observed with step size 0.1, the level of consensus (indicated by the variance among the agents) is quite unstable. On the other hand, with very low step size 0.001, the level of consensus is quite stable (moving average of variance remains 0). However,

the convergence is extremely slow. This observation conforms to the theoretical analysis described in the paper as well as justifies the choice of step size $0.01$ in the experiments presented above.