[Reviews · NeurIPS 2017]

Reviewer 1



This paper proposes a variant of stochastic gradient decent for highly decentralized networks where communication is only possible through "neighbors" is a distributed device graph. While this computation model is not realistic for conventional deep learning systems, it addresses the interesting case of trying to learn using a highly distributed network of weak devices, such as cell phones or IOT devices. This is a setting where distributed methods are not as well studied as in the centralized case, and I think the proposed class of algorithms is more practical in this setting than, say, distributed asynchronous ADMM methods that require updates of dual parameters. The Lyapunov function analysis presented is fairly elegant and concise. One criticism I do have of this paper is that convergence requires a fairly complex and uninterpretable assumption (Assumption 3) that seems fairly non-standard for SGD analysis. Also, It would be nice to see some experiments on convex models (like logistic regression), where the assumptions are verifiable.

Reviewer 2



This paper considers the problem of distributed optimization on networks where machines can only directly communicate with neighboring machines in rounds, as specified by a weighted incidence matrix. The method they propose is a version of the algorithm from Nedic and Ozdaglar [17], with the deterministic gradients replaced by stochastic ones. The authors do not clearly explain this in their related work, just stating that “the proposed algorithms have similarities”. As it currently is, I'm not happy with the statement of the algorithm. It is not sufficiently detailed for me to know exactly what is intended. In particular, the quantity g_j is referred to as a stochastic gradient without any further elaboration. My best guess is that it's intended to be chosen via without-replacement sampling, as they refer to shuffling the data in their pseudo-code. I think the proof assumes that the data are sampled without-replacement though (please clarify if this is the case). In either case, the pseudo-code of the algorithm must make this clear. The momentum variant is interesting, I think the paper would be much stronger if a theoretical analysis of it was included.

Reviewer 3



This paper explores a fixed peer-to-peer communication topology without parameter server. To demonstrate convergence, it shows that the Lyaounov functions that is minimized includes a regularizer term that incorporates the topology of the network. This leads to convergence rate bounds in the convex setting and convergence guarantees in the non-convex setting. This is original work of high technical quality, well positioned with a clear introduction. It is very rare to see proper convergence bounds in such a complex parallelization setting, the key to the proof is really neat (I did not check all the details). As we know, insight from convex-case proofs usually apply to the non-convex case, especially as the end convergence should be approximately convex. Thanks to the proof, one can predict the impact of the topology over convergence (through the eigenvalues). This proof should also open to further work. Experiments are also well designed, thought they do not cover the hypothetical situation where a parameter server could fail. However, several critical limitations will impact the short term significance. Some of them could probably be addressed by the time of publication. First, the loss in both training speed and convergence accuracy of CDSGD compared to plain SGD is very large, and is only mitigated with the use of a momentum. While the authors show how the algorithm can be modified to support both Polyak and Nesterov momentums, I could not find anywhere which momentum CDMSGD actually use! There is no explanation of why this momentum helps performance so much. Second, compared to other data-distributed SGD approaches such as Federated Averaging, the main practical contribution of the paper is to get rid of the parameter server, at some efficiency cost on controlled experiments (figure 2.b). So this work significance would greatly improve if the authors clearly identified the reasons for not using a parameter server. In the introduction, they hint at better privacy preserving (if I understood right), but Federated Averaging also claims privacy preserving WITH a parameter server. In the experiments, final accuracy with momentum is better than Federated averaging: is this because of the decentralized approach or the use of a momentum? Note also that this paper does not seem to mention the main reason I have seen for using decentralized approach, which is robustness in case of failure of the central server. Last, the stochastic Lyapunov gradient (eq 7) shows that the regularization can be very strong especially with small learning rates, and the effects are visible in the experiments. What this regularization amounts to would be worth studying, as this is a striking side-effect of the algorithm. Potential issues: Line 92: f_j(x)=1/N f(x) does not make sense to me. It should be f_j(x)=1/N sum_i f^i_j(x) ?? Line 119: it implies that all eigenvalues are non-zero, which contradicts the fully connected topology where they are all zero but one. Line 143: "Gradient of V(x) also has a Lipshitz": you either remove gradient or replace 'has' with 'is'.